# Telemedicine for Pre-Employment Medical Examinations and Follow-Up Visits on Board Ships: A Narrative Review on the Feasibility

**DOI:** 10.3390/healthcare9010069

**Published:** 2021-01-13

**Authors:** Getu Gamo Sagaro, Marzio Di Canio, Emanuele Talevi, Francesco Amenta

**Affiliations:** 1Telemedicine and Telepharmacy Center, School of Medicinal and Health Products Sciences, University of Camerino, 62032 Camerino, Italy; francesco.amenta@unicam.it; 2Research Department, Centro Internazionale Radio Medico (C.I.R.M.), 00144 Rome, Italy; mdicanio@cirmservizi.it (M.D.C.); etalevi@cirmservizi.it (E.T.)

**Keywords:** telemedicine, seafarers, cardiology, healthcare, technology

## Abstract

Background: Telemedicine has already been applied to various medical specialties for diagnosis, treatment, and follow-up visits for the general population. Telemedicine has also proven effective by providing advice, diagnosis, and treatment to seafarers during emergency medical events onboard ships. However, it has not yet been applied for pre-employment medical examinations and follow-up visits on board ships. Objective: This review aimed to assess the possibility of using telemedicine during periodic visits between one pre-employment medical examination and others on board ships, and to recommend necessary medical examination tests with screening intervals for seafarers. Methods: Various databases including PubMed, EMBASE, Scopus, CINAHL, and Cochrane Library were explored using different keywords, titles, and abstracts. Studies published between 1999 and 2019, in English, in peer-reviewed journal articles, and that are conference proceedings were considered. Finally, the studies included in this review were chosen on the basis of the eligibility criteria. Results: Out of a total of 168 studies, 85 studies were kept for further analysis after removing the duplicates. A further independent screening based on the inclusion and exclusion criteria resulted in the withdrawal of 51 studies that were not further considered for our analysis. Finally, 32 studies were left, which were critically reviewed. Out of 32 accepted studies, 10 studies demonstrated the effectiveness of the electrocardiogram (ECG) in monitoring and managing remote patients with heart failure, early diagnosis, and postoperative screening. In 15 studies, telespirometry was found to be effective in diagnosing and ruling out diseases, detecting lung abnormalities, and managing patients with chronic obstructive pulmonary disease (COPD) and asthma. Seven studies reported that telenephrology was effective, precise, accurate, and usable by non-medical personnel and that it reduced sample analysis times and procedures in laboratories. Conclusion: using new technologies such as high-speed internet, video conferencing, and digital examination, personnel are able to make the necessary tests and perform virtual medical examination on board ships with necessary training.

## 1. Introduction

Telemedicine is the delivery of medical services over distance through information and communication technologies [1]. Telemedicine applications include disease prevention and control, healthcare, health promotion, education, training, and research for health [1]. Telemedicine can be traced back to the end of the 19th century to the mid-20th century with the use of radiotelegraphy to provide medical advice to patients onboard ships [2]. The evolution versus modern telemedicine began in the 1960s, mainly spurred on by the military and space technologies, as well as by some people using readily available commercial equipment [3,4]. More recently, the applications of telemedicine are manifold and involve both developed and developing countries, rural areas, islands, and in all situations where distance could limit or make difficult the provision of medical care [5].

Merchant ships represent one of these potential scenarios as they spend long periods on the high seas. Most of the time, merchant ships do not carry a doctor or other health professional on board, and the healthcare is in the hands of the captain or his delegate when seafarers are injured or ill [2,6]. Seafaring is a hazardous profession when compared to land workers [7]. This is probably due to the intrinsic nature of the work rich in psychophysical stressors, including isolation from the family, the multiculturalism of the crews, limited recreational opportunities, and long work shifts [8]. Seafarers exhibit high mortality and morbidity when compared with workers onshore for both the deck and machine sectors [9]. They are also prone to unhealthy lifestyles, such as smoking and unhealthy diets [10]. These risk factors, combined with chemical exposure, make seafarers one of the categories with the highest rate of certain cancer and cardiovascular diseases (CVD) [8]. The most frequent causes of death are attributable to shipboard accidents, work-related accidents, CVD, and poisonings [11,12].

Occupational medicine is an internationally recognized branch of preventive medicine with a large diffusion in all EU countries with the broad spectrum of its influence. One of the main objectives of occupational medicine is the assessment of suitability or unsuitability for a given work, temporary or permanent. Moreover, occupational medicine has the task of assessing the risks associated with the workplace to protect and promote people’s health by preventing diseases and accidents that could occur in the workplace [13]. In terms of telemedicine applications to occupational medicine, more than 50% of hospitals in the USA are using telemedicine to treat their patients and over 74% of companies offer telemedicine services to their employees as a part of their healthcare programs [14,15]. Moreover, in the UK, telemedicine is widely used and booming [16]. Telemedicine is often used in postoperative clinical programs and in the follow-up of patients with chronic diseases such as diabetes [17]. Telemedicine is widely demonstrated in the literature in many medical specialties such as pediatrics, neurology, cardiology, diabetology, psychiatry, postoperative orthopedic care, primary care, and emergency medicine in cognitive disorders [18,19,20,21,22,23,24,25]. However, only a few studies have focused on the use of telemedicine in maritime occupational medicine, especially virtual medical examination/test application during emergency on board ships.

The World Labour Organization (WLO) and the International Maritime Organization (IMO), in the report “Guidelines on the medical examination of seafarers”, published recommendations concerning pre-employment medical examinations (PEME). The purpose of the PEME is to reduce the risk of illnesses or injuries to the seafarer on board. These visits are carried out every two years and involve certain tests proposed by the WLO and the IMO [26]. Protocols for pre-employment medical examination developed by national maritime authorities or other insurers, agents, or private companies may change screening intervals of less than the expected two years, some of which are carried out directly on board ships [10].

In Italy, the regulations in force provide that a competent doctor, appointed by the shipowner, within the framework of a health protocol, adequately monitors the health conditions of seafarers by carrying out periodic visits. Different medical tests are taken into account during each visit (health check at the time of recruitment/first visit and periodic check-ups), and medical examinations are considered, such as blood chemistry tests, electrocardiogram (ECG), visual function examination, audiometry, and other tests on doctor’s recommendation, particularly during periodic visits. The legislation requires the competent doctor to collaborate with the employer/shipowner and/or the captain and with his prevention and protection service in order to carry out preventive and periodic health checks, expressing suitability for the specific task. The competent (occupational) doctor is also called upon to establish the health records of the individual worker, communicating any aptitude in writing to the employer/owner and to the worker [27,28,29]. However, there are lengthy processes for a medical check-up before boarding services; shipowners’ requests are addressed to the SASN (Servizi Assistenza Sanitaria Naviganti)-Health Assistance Services for Navigation, Maritime and Civil Aviation Personnel, reference USMAF-SASN for the territory or to an authorized medical officer [30]. After approval, depending on the appointment, periodic medical check-ups are to be carried out, with most medical examinations being performed onboard ships by a doctor or other health professionals. Hence, telemedicine for periodic medical check-ups can reduce unnecessary travel, skip/shorten some process steps, reduce transport costs for regular check-ups, avoid waiting lists, and increase the quality of services by offering specialty doctors the opportunity to participate in virtual examinations.

Telemedicine has proven effective by providing advice, diagnosis, and treatment to seafarers during emergency on board ships [31]. It has already been applied to various medical specialties for diagnosis, treatment, and follow-up visits for the general population and has made it possible to overcome various constraints such as geography and the resources to provide healthcare to remote populations [32]. However, it has not yet been used for pre-employment medical examinations and follow-up visits onboard ships. The purpose of the present review was to assess the possibility of using telemedicine during periodic visits between the pre-employment medical examination and others and to propose necessary medical tests with screening intervals to be included in PEME protocols.

## 2. Materials and Methods

The review was conducted by searching the different published scientific literature, indexed in various databases, including PubMed, EMBASE, Scopus, CINAHL, and Cochrane Library. We included studies in areas of 4 medical specialties reporting medical exams/tests frequently used in telemedicine, namely, electrocardiogram (ECG), spirometry, blood glucose monitoring, and urinalysis. Different key terms for search, including “telecardiology”, “tele diabetology”, “telespirometry”, “telenephrology”, “teleconsultation”, “telemedicine”, and “telehealth” were used. Inclusion criteria for selected studies included studies published between 1999 and 2019, studies published in English, peer-reviewed journal articles, conference proceedings, and full-text paper. Studies published with only abstracts and not in English were excluded.

In total, 168 potentially relevant studies were selected from the above databases. Out of a total of 168 relevant studies, 85 studies remained after the duplicates were removed. A further 51 studies were excluded upon reviewing the abstract, title, and assessment of full-text as not consistent with our research questions. Finally, 32 studies were selected after a thorough review and included in this review (Figure 1). The review analysis involved 3 independent reviewers and an expert in the event of disagreement.

The summary of results for the application of telemedicine in various medical examinations/tests of occupational medicine interest is summarized in Table 1 (Table 1).

## 3. Telemedicine in Different Medical Examinations/Tests

### 3.1. Electrocardiogram (ECG)

Telemedicine is not a separate medical modality but includes a growing variety of applications and services that use telephone lines, videos, e-mails, smartphones, wireless tools, and other forms of telecommunication technology. Among the wide range of medical specialties in which telemedicine has been successfully applied, cardiology has been found to be one of the most common fields of application. Through the transmission of clinical data and the electrocardiogram (ECG), telecardiology allows access to a real-time assessment (teleconsultation) without the need to travel for both the patient and the cardiologist. Telecardiology has proven to be useful in the clinical management of remote patients with real or suspected heart disease in different clinical settings. Over 20 years, several attempts have been made to try to introduce and expand telecardiology in the hospital setting, especially for the diagnosis and treatment of patients in remote locations [33]. Positive impacts of telecardiology have been demonstrated in the literature on patients with heart failure. In these patients, remote monitoring compared to traditional monitoring procedures decreased the risk of recurrence of the event [34].

A study conducted in 2010 demonstrated that the accuracy of an ECG performed remotely via wireless procedures in detecting an episode of atrial fibrillation was 93%, and the accuracy of this examination was 94% [35]. A study by Ong et al. (2016) focused on the effectiveness of remote patient monitoring (RPM) for the transition of care in adult patients with heart failure, finding no significant difference in patient readmissions for 180 days. Those who received the remote monitoring intervention had readmission rates of 50.8%, while those who did not receive the intervention had readmission rates of 49.2% [36]. Another study compared hospital readmission rates and death rates for two groups of heart failure patients, followed by telemonitoring and conventional monitoring. The types of post-treatment screening were comparable in that no statistically significant difference was found. Readmission rates were 49.3% for remote monitoring and 47.4% for routine care, while the mortality rate for remote monitoring was 11.1%, and in routine care it was 11.4% [37].

The use of portable, non-invasive tele-cardiologic screening equipment was found to be less costly for hospitals and more comfortable for patients, allowing them to remain in domestic environments [38]. According to Sohn, costs would be around 25% lower in patients with mild symptoms [39]. In addition, to diagnosis and postoperative screening, telecardiology can also be used in prevention. In 2010, an Italian project was launched aimed at using a telecardiology device to perform early diagnosis of 13,016 students aged 16 to 19 [40]. In 2016, the same author collected the results showing that 24% of the suspects had altered signals in the electrocardiogram. The conclusion was that the use of telecardiology procedures in mass screening has many advantages such as lower costs, the possibility of use in environments far from hospitals, and not requring qualified personnel [41]. Another study evaluated the effectiveness of remote monitoring through the use of various devices, including the electrocardiogram (ECG) in healthy clients, in patients at risk for cardiovascular diseases such as those with diabetes and hypertension, and in patients with a previous cardiovascular event. The healthy clients were monitored by a 12-lead ECG installed in pharmacies and connected to a telemedicine platform manned by a cardiologist 24 h a day. Between 1 January 2015 and 31 December 2017, the study involved a total of 79,898 women (mean age 52 ± 12 years) and 68,458 men (mean age 49 ± 11 years). Of all ECGs performed, approximately 8% showed electrocardiographic abnormalities inconsistent with the patient’s medical history. The authors confirm that tele-cardiological screening systems can be used successfully in the prevention of cardiovascular diseases with a consequent positive impact on public health [42].

### 3.2. Spirometry

Spirometry, also defined as spirometric test or simply respiratory function test, is a diagnostic test that is performed using a spirometer, a computerized instrument, connected to a mouthpiece. It is a very simple, painless, and non-invasive examination. Spirometry plays an important role in the diagnosis and monitoring of chronic obstructive respiratory disease. It should be noted that relevant clinical guidelines indicate the need for widespread use of spirometry in primary care for the early diagnosis and appropriate management of chronic asthma and chronic obstructive bronchopulmonary disease (COPD). Hence, this test is of relevant importance in the health surveillance in preventive medicine for seafarers who, for working reasons, often need to face long journeys [43,44]. The possibility of remote use of spirometry through telemedicine equipment has been investigated by several authors both as a prevention and monitoring tool in patients suffering from respiratory syndrome [45]. Telespirometry is used in clinical practice to monitor asthma patients and patients with COPD living at a distance from the hospital [46,47]. These integrated systems, in patients with respiratory diseases and frequent exacerbations, can reduce both emergency room visits and the number of hospitalizations [48].

In 2009, Bonavia published an article reporting an Italian project investigating the possibility of using telespirometry in general medicine. As a part of the project, 937 family physicians exchanged data via telespirometry equipment with 56 specialist centers, visiting their patients with risk factors, persistent respiratory symptoms, or a previous diagnosis of asthma or chronic obstructive pulmonary disease for two years. About 90% of the spirometry tests (20,757 spirometry tests performed in total) met the criteria and made it possible to make a diagnosis or rule out pathologies. 40% of the spirometries made it possible to detect pulmonary abnormalities. The authors consider telespirometry to be a reliable and useful alternative in the management by general practitioners of chronic respiratory diseases. The quality of the spirometric examination is highly dependent on the skills of the technician administering the test. The pulmonology company’s guidelines indicate the skills and training to be acquired to manage this exam [49]. An interesting perspective, which involves several studies, is given by the possibility of remote monitoring at home. This technology enables the self-measurement of clinical parameters/symptoms of patients at home and allows the communication between healthcare professionals and remote patients [50,51].

A study investigated the possibility of carrying out the spirometric test directly at home in total autonomy. The study involved four patients with previous chronic obstructive bronchopathy who were provided with telespirometry equipment equipped with tablets capable of supporting the patient in the examination. This system was used for 12 weeks in which patients performed several daily spirometries without any medical assistance. As a result, the large part of the spirometry (94.5%) was considered acceptable and usable by qualified personnel [52]. Another study investigated the possibility of self-administration of the spirometric examination in asthmatic patients [53]. The author equipped 86 asthmatic patients with a portable instrument for spirometric measurements at home without medical supervision, evaluating their acceptability according to the American Thoracic Society and European Respiratory Society (ATS/ERS) criteria. The author set the primary endpoint with the following criteria: correct use of the device three or more times within 7 days (±1 day) in one of the 3 weeks of the study. Of 78 patients, 67 (86%) reached the primary endpoint. Seventy-five (96%) participants used the device correctly one or more times, and 10 (13%) patients managed to use the tool every day during the 3 weeks. The authors showed that remote self-assessment using spirometry equipment is a feasible practice [53].

Adequate training in performing the spirometric examination of technicians or operators onboard the vessel capable of ensuring high-quality standards in spirometry could be fundamental to generate reliable results that the occupational physician could then assess from another clinic suitable for the examination, obviously using portable technological spirometers. The quality of spirometry tests strongly depends on adherence to international recommendations [53,54]. There are different tools today, such as those that use Android micro-control technology to measure lung function. For example, this latest technology has reported excellent results on patients who have been analyzed both with an examination performed with the traditional spirometer and with the new micro-control technology providing very similar final values, with an accuracy of high measurement and for forced expiratory volume in 1 s (FEV1) and forced vital capacity (FVC) indicating that this device, for example, could be usefully used in telemedicine and health surveillance. Furthermore, there is a remote assistance technology that uses a simple spirometer with a Bluetooth module, an ES application based on MATLAB, and a mobile app based on Android. In this case, the portable spirometer used in this study can be connected to a mobile phone via Bluetooth. This technology has been used to evaluate the chronic course of diseases such as COPD and asthma. During 6 months, 780 patients were assessed and diagnosed with an accuracy of 97.32% [55].

### 3.3. Blood Glucose Monitoring

Telemedicine interventions for diabetes can vary from simple reminder systems via text messages over complex web interfaces. Patients can upload their glucose levels measured with a home meter and other relevant data such as medications, eating habits, level of activity, and anamnesis. The measurement of blood glucose concentration represents an essential step in the management of diabetic disease—glycemic self-monitoring, in fact, is widely used by patients with type 1 and type 2 diabetes to verify metabolic compensation, to identify and treat episodes of hyper- and hypoglycemia and to adapt the hypoglycemic therapy to the conditions of life (nutrition, physical activity, stress, intercurrent diseases) [56]. In the field of diabetes, telemedicine is used in its various forms: remote glycemic monitoring, teleconsultation, personal medical records, telenursing, and call centers. Several studies have shown that real-time transmission of blood glucose data is achievable with evidence demonstrating improvements in terms of glycemic control [57].

### 3.4. Urine Analysis

Urinalysis is an important diagnostic screening test useful for diagnosing and monitoring nephrological and urological conditions. This test is also often used in general preventive screening [58]. Information technology has significantly reduced the analysis times following the collection of urine and often performed in specialized laboratories. Nowadays, many portable electronic readers are available that can analyze in real-time the urine collected on particular test strips [59]. To be effective, these devices must be economical, portable, precise, reliable, robust, powered by batteries, and usable by non-medical personnel as well [59].

A recent development consists of the use of smartphones to read and interpret the results of the test strips where urine is collected [60]. These smartphones are often combined with electronic pocket readers and reagent strips [61]. Dae-Sik Lee proposed in 2011 a mobile health platform by combining a pocket colorimetric reader with a smartphone and paper strips for urinalysis capable of analyzing glucose, proteins, bilirubin, ketones, nitrites, pH, specific gravity, erythrocytes, and leukocytes. Various tests carried out show that the device is efficient, inexpensive, and accurate [62]. The urine test strip is a device made up of strips on different distinct reactive zones allowing for the determination of specific gravity, pH, proteins, glucose, ketones, bilirubin, blood, nitrites, urobilinogen, and leukocytes in the urine. Often, the results have to be interpreted, and, in order to avoid human error, some digital scanners are able to interpret the results independently via the smartphone camera [63,64].

## 4. Legal Implications of Telemedicine

Telemedicine has significant repercussions in the delicate ethical sphere, as this different way of managing the interaction and communication between the patient and the doctor (or in general, the health workers involved) impacts a particular situation for who is in need of health care. On the other hand, on establishing the patient–doctor relationship, the safeguard of the dignity of the patient should be always safeguarded.

The legal problems related to telemedicine have not yet found standard solutions at the international level. Any request for telemedicine is considered a medical act. Hence, although many teleconsultation procedures are unique, the traditional principles of traditional doctor–patient relationships are also valid in telemedicine. Three concerns can be involved in legal performance issues [65,66]. As a result, in terms of the person who transmits the data, “informed consent must govern the relationship between the patient and other interested parties.” This should include the patient’s awareness of the technical aspects; the potential risks; the precautions required; and, at the same time, and the guarantee of the confidentiality of information [66]. The person who receives the data is the medical service user. Regarding the service provider, confidentiality, as well as the quality of the transmitted and received data, must be guaranteed by the service provider [66].

## 5. Recommendations

Telemedicine can be used successfully for pre-employment medical examinations and follow-up visits on board ships. This study used a narrative review to evaluate the possibility of telemedicine for periodically pre-employment medical exams and follow-up visits for workers needing to be followed aboard ships. We considered four major medical exams/tests for telemedicine applications in this review, namely, electrocardiogram (ECG), spirometry test, blood glucose monitoring, and urinalysis. As a result of telemedicine, we proposed the necessary medical examinations/tests with screening intervals to be included in the pre-employment medical examination (PEME) protocols (see Table 2). Table 2 summarizes features, medical tests, suggested periodicity, telemedicine modality, and equipment to be used for telemedicine practice in occupational medicine. Possible applications of telemedicine in the PEME, follow-up, and future visits are detailed below:PEME: ECG, spirometry, measurement of vital signs (blood pressure, heart rate, and temperature), oxygen saturation assessed by a pulse oximeter, blood glucose measurement by glucometer, and urinalysis are tests that can be monitored remotely in some cases even without medical assistance. These tests are currently used for the general population in different medical specialties. We thus suggest using them both for PEME visits and/or for periodic checks of seafarers aboard ships, provided that on board there is the necessary technological resource training.Follow-up visits: Telemedical Maritime Assistance Services (TMAS) doctors can make multiple medical visits a single patient, guaranteeing appropriate evaluation standards. They can also follow the same patient for the necessary time by monitoring over time without the need for large displacements and without the need to visit inside the ships in person. The technological equipment should provide a secure and high-speed internet connection; a clinical telemedicine software that acts as a hub capable of sending the patient’s vital parameters to the doctor; devices capable of monitoring the patient’s body parameters; and, finally, an ad hoc training program with periodic simulations. If fitted onboard, these systems could also help a TMAS doctor make a correct diagnosis and plan adequate treatment.Future activities: the possibility of carrying out preventive medicine tests using telemedicine technologies could be considered in the future as a fundamental element for remote maritime preventive medicine practice. Specific experimental studies of devices integrated into platforms, protocols, and patient satisfaction should be conducted, possibly using targeted comparisons with traditional systems.

## 6. Conclusions

This narrative review has highlighted how telemedicine and devices for the detection of body parameters in the medical field are extensively practiced in many specialties of modern medicine. This review evaluated telemedicine’s feasibility for pre-employment medical examinations (PEMEs) and follow-up visits aboard ships. It recommended the necessary medical examinations/tests such as ECG, spirometry test, blood glucose monitoring, and urinalysis with screening intervals for seafarers’ onboard ships be considered in the PEME protocol. Moreover, recommendations were provided to responsible bodies, stakeholders, and researchers to implement telemedicine during the PEME and study its effectiveness. However, in many countries around the world, telemedicine is not an integral part of the health system, and thus it is legitimate to assume that the full potential of telemedicine has not yet been exploited. This review shows that with the advent of new technologies such as high-speed internet, video conferencing, and digital examinations, it is possible to report the necessary tests and perform virtual medical exams on board vessels. Telemedicine can be a fundamental element for the prevention and treatment required in health surveillance, particularly for the conditions in which reaching the patient is difficult and expensive, such as onboard ships.

## Figures and Tables

**Figure 1 healthcare-09-00069-f001:**
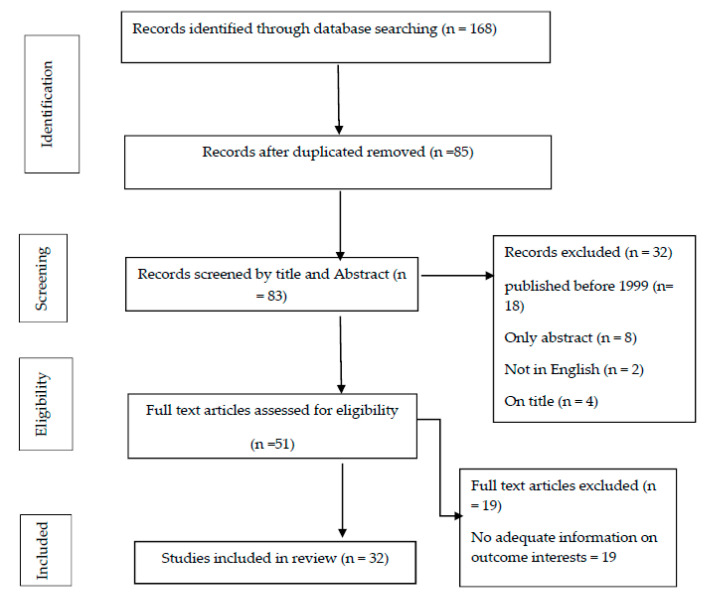
Literature search flowchart with eligibility criteria.

**Table 1 healthcare-09-00069-t001:** Summary of results for the application of telemedicine in different medical examinations/tests.

Area of Application	Medical Examination/Test	Purpose of Application	Outcome	Reference(s)
Telecardiology	Electrocardiogram (ECG)	 For tele-transmission and teleconsultation in the management of remote patients with heart failure  Telemonitoring and postoperative screening of patients with cardiovascular cases  Prevention screening	✓Reduction of costs for diagnosis; ✓Accurate in detecting an episode of atrial fibrillation (93%) and examination (94%).✓Effective remote patient monitoring, including early diagnosis and postoperative screening.	Molinari G. et al. [33]Inglis S.C. et al. [34]Lin C.T. et al. [35]Ong M.K. et al. [36]Herrin J. et al. [37]Majumder S. et al. [38]Sohn S. et al. [39]De Lazzari C. et al. [40]De Lazzari C. et al. [41]Lupi, L. et al. [42]
Telespirometry	Spirometric test	 Respiratory function test  For early diagnosis and management of patients with COPD, asthma	✓Effective by diagnosing and rule out pathologies, detecting pulmonary abnormalities.✓Successful self-management of clinical parameters of patients at home.✓Effective in managing patients with COPD and asthma (97.32% diagnosing accuracy).	Kim et al. [43]Zealand N. [44]Molina-bastos C.G. et al. [45]Averame G. et al. [46]Toledo P.D. et al. [47]Owens M. et al. [48]Jouneau S. et al. [49]Vitacca M. et al. [50]Ohberg F. et al. [51]Kupczyk M. et al. [52]Redlich C.A. et al. [53]Stout J.W. et al. [54]Fung A.G. et al. [55]
Tele-diabetology	Blood glucose monitoring	 Remote glycemic self-monitoring.  Telenursing, personal medical records, and tele-transmission.	✓Effective in diagnosis, treatment, self-glycemic monitoring, and follow up.	Bruttomesso D. et al. [56]Rodriguez-Gutierrez R. et al. [57]
Telenephrology	Urinalysis	 Diagnosis and monitoring of the subject’s nephrological and urological conditions.  Prevention screening.	✓Reduction of time of sample analysis and procedures in laboratories.✓Effective, precise, accurate, and usable by non-medical personnel.✓Successful in analyzing various tests, including glucose, proteins, bilirubin, ketones, nitrites, pH, specific gravity, erythrocytes, and leukocytes.✓The device is efficient and inexpensive.	Hannemann-Pohl K. et al. [58]Langlois M.R. et al. [59]Mohammadi S. et al. [60]Ginardi R.V.H. et al. [61]Lee D.S. et al. [62]Soldat D.J. et al. [63]Montangero M [64]

**Table 2 healthcare-09-00069-t002:** The proposed general medical examinations and tests for seafarers on board ships using telemedicine technologies.

Features	Medical Examination	Periodicity	Telemedicine Modality	Equipment
**General well-being**	Medical history	Annual basis	Video, e-mail	Telemedicine devices
**Noise**	Audiometry test	Annual basis	Audio, video	Telemedical devices
**General well-being** **Stress-related**	Electrocardiogram (ECG), spirometry, urinalysis	Every six months	Still image Video	Telemedical devices
**Vibrations**	Postural evaluation: manual handling of loads	Annual basis	Video	Telemedicine devices
**Exposure to toxic substances**	Blood chemistry tests aimed at checking cardiovascular and renal function (blood count, blood sugar, liver and kidney function, lipid structure); leukocyte formula, urinalysis, determination of urinary hydroxypyrene.	Annual basis	E-mail, audio, video	Telemedicine devices
	Pulse oximetry, blood pressure	Every month	Video, e-mail	Telemedical devices
**Postural workload**	Body mass index evaluation; postural evaluation	Annual basis	Video	Telemedical devices
**Fatigue and stress assessment**	Electrocardiogram and saturimetry for fatigue; stress assessment, see the appropriate section of this protocol	Annual basis	Video, e-mail	Telemedical devices

## Data Availability

Data sharing not applicable.

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
