# Peer review of "Telemedicine for Pre-Employment Medical Examinations and Follow-Up Visits on Board Ships: A Narrative Review on the Feasibility"

_healthcare, 2021, doi:10.3390/healthcare9010069_

Round 1
Reviewer 1 Report
Dear authors.
The papers intends to present a review or survey about the telemedicine for pre-employment medical examinations.
Some minor corrections about your paper:
-References are required in the first part of the introduction (What is telemedicine?).
-References format does not fit with journal style -In the introduction are missed the contributions and the structure of the paper. -Table 1 should be located as near as possible of the end of section 2 (Methods), where is cited, not near to the end of section 3. -To be a review, there are many older references (2005 - 2013) Unfortunately, I guess the paper is a little bit poor trying to expose a complete survey or review about the topic, because does not address as far as it should be to present a "sound" research about this topic. In addition, the problem or context is not introduced well, and the importance of the topic is not clear to the reader.Author Response
Dear reviewer, we thank you for taking the time to read our paper carefully and for the valuable comments and suggestions you have provided, which helped us improve the revised manuscript that we are re-submitting for review.
Please find below our response to each of the comments and suggestions:
With best regards!
Getu Gamo Sagaro (Ph.D.)
On behalf of all authors
Reviewer #1
- References are required in the first part of the introduction (What is telemedicine?).
We have included the reference in the revised version of the manuscript.
- References format does not fit with journal style
To address this comment, the journal has no specific reference style, but we have tried to meet the journal's recommended reference style in our revised version of the manuscript.
- Table 1 should be located as near as possible of the end of section 2 (Methods), where is cited, not near to the end of section 3.
We thank the reviewer for his concern and the change has been made in the revised version according to the reviewer’s comment.
- To be a review, there are many older references (2005 - 2013) Unfortunately, I guess the paper is a little bit poor trying to expose a complete survey or review about the topic, because does not address as far as it should be to present a "sound" research about this topic.
To address these comments, as we mentioned in the materials and methods section, we considered for this review, the studies published between 1999 and 2019. Thus, some studies are old, as the reviewer mentioned, but we included them per our eligibility criteria. Secondly, although telemedicine technology started during the 19th century, it is still not well applied in everywhere and understudied areas, especially offshore. Hence, we used a narrative review for this study, and based on the studies conducted on the general population, we assessed seafarers' possibility to apply during periodic visits. We recommended that some medical tests be included in the PEME protocol. This is the first review, and it would be the baseline for future studies.
- In addition, the problem or context is not introduced well, and the importance of the topic is not clear to the reader.
To address these comments, we accepted the comments, and the changes have been made in the revised version. Regarding the topic's problem or context, revised under introduction page number 2, L82 - 86 and in the same page and page 3 from L90 – 94. Regarding the significance of this review revised under introduction page number 3, L 95 – 103.
Reviewer 2 Report
1) Introduction: there is a large part on seafarers, but is this really a relevant group of patients? How many people are seafarers? Perhaps this section should be shortened.
2) Introduction LL 64-66: How many hospitals are using telemedicine? 20 or 50%
3) A figure showing how many studies were included and excluded could be helpful.
4) In my opinion, the authors could improve the manuscript by discussing the topic telemedicine more controversial: besides legal aspects, what are the disadvantages of telemedicine? What about social contacts of the patients? Can examinations and data be manipulated? What is the role of telemedicine in the pandemic?
Author Response
- Introduction: there is a large part on seafarers, but is this really a relevant group of patients? How many people are seafarers? Perhaps this section should be shortened.
To address these comments, of course, in the introduction, more about seafarers. The reason why we assessed the possibility of telemedicine for seafarers during pre-employment medical examinations/tests. Seafarers are seaman or mariners working onboard ships, and currently, more than 1.6 million seafarers are on duty at sea, and approximately 20 - 25 seafarers are per ship. They are working in hazardous and isolated environments and providing healthcare for these groups is not as easy as employees working on the land. We appreciate the reviewer for understanding why we have focused more on seafarers.
- Introduction LL 64-66: How many hospitals are using telemedicine? 20 or 50%
We thank the reviewer for his invaluable comments, and we have revised the sentence in the revised version of the manuscript in introduction L62- 64.
- A figure showing how many studies were included and excluded could be helpful.
The comment is accepted, and the figure showing the literature search flowchart with inclusion and exclusion criteria is included in the revised version of the manuscript on page 4.
- In my opinion, the authors could improve the manuscript by discussing the topic of telemedicine more controversial: besides legal aspects, what are the disadvantages of telemedicine? What about social contacts of the patients? Can examinations and data be manipulated? What is the role of telemedicine in the pandemic?
To address these comments, in the manuscript, we have already included the legal aspects of telemedicine. We have clearly mentioned the importance of this topic and the rationale in the introduction. Also, on recommendations, we have tried to point out the benefits of this technology, especially for seafarers. We did not mention the role of telemedicine in the pandemic because this is out of our scope of review. We thank the reviewer for understanding the scope of this review.
Reviewer 3 Report
It is with great pleasure that I participate in the review of this article.
Why do you call this review “narrative review” in the title? In my opinion, having no associated method is a literature review.
This article would be much more interesting and valuable if it were converted into a systematic review of the literature. As a reviewer, and since this work is exempt from the scientific method, I will only limit my comments to the structure and content presented.
Section 3 - They should write that this section already derives from the analysis of the identified articles.
In Table 1 - References should be associated with the results and not in a very general way with the line. In the text, the findings of the articles are described in more detail. Here the most relevant results of the studies could be identified with the appropriate bibliographic reference (as such it suggested the elimination of the column: "references(s)".
In Table 2, you could follow the same methodology (Table 1), identify which are studies on which they are based these proposed. Otherwise, this are the opinion of the authors.
Author Response
- Why do you call this review “narrative review” in the title? In my opinion, having no associated method is a literature review.
To address this comment, our reviewer is the first review regarding PEME onboard ships, and telemedicine technology has not yet been used for PEME so far. We, therefore, evaluated some studies conducted on the general population or on employees who are working ashore and evaluated based on these studies the possibility of applying telemedicine to seafarers at sea during periodic medical check-ups. In other words, a general discussion of a topical approach to feasibility assessment.
- This article would be much more interesting and valuable if it were converted into a systematic review of the literature. As a reviewer, and since this work is exempt from the scientific method, I will only limit my comments to the structure and content presented. Section 3 - They should write that this section already derives from the analysis of the identified articles.
To answer these comments, we have already mentioned why we used the narrative review rather than systematic under the reviewer’s first point. As for section 3, we have clearly cited the studies we used for the analysis under each sub-heading of section 3, and it obviously derived from the analysis of selected studies.
- In Table 1 - References should be associated with the results and not in a very general way with the line. In the text, the findings of the articles are described in more detail. Here the most relevant results of the studies could be identified with the appropriate bibliographic reference (as such it suggested the elimination of the column: "references(s)".
To address these comments, we summarized the results of selected studies on each outcome interest and accordingly cited the references. Hence, we do not think that it is more detailed rather than indicating the aims of telemedicine application and its outcome by each medical examination/test.
- In Table 2, you could follow the same methodology (Table 1), identify which are studies on which they are based these proposed. Otherwise, these are the opinion of the authors.
In Table 2, we have recommended or proposed the necessary medical examinations/tests with screening intervals for seafarers’ onboard ships to be considered in the PEME protocol based on past experiences. Some medical examinations/tests are being carried out on board ships for seafarers upon the shipowner’s requests, but not organized formally and included in the PEME protocol. In general, these proposed medical tests are the authors' opinions, and among us, two authors are medical doctors who are currently providing healthcare services for seafarers.
Round 2
Reviewer 3 Report
I thank you in advance for the responses and the changes made. It is unfortunate that it is not a systematic review, it would have another more significant impact. But, even so, it is the first publication on the subject.
My congratulations for that.